# Determinants of anthroponotic cutaneous leishmaniasis by case-control study in Morocco

**Mounia Amane**[1], **Mohamed Echchakery**[1,2], **Mohamed Daoudi**[1], **Mohamed Hafidi**[1], **Samia Boussaa**[1,3]*

**1** Microbial Biotechnologies, Agrosciences and Environment Laboratory (BioMAgE), Faculty of Sciences Semlalia, Cadi Ayyad University, Marrakesh, Morocco, **2** Epidemiology and Biomedical Unit, Laboratory of Sciences and Health Technologies, Higher Institute of Health Sciences, Hassan First University, Settat, Morocco, **3** ISPITS-Higher Institute of Nursing and Technical Health Occupations, Ministry of Health and Social Protection, Rabat, Morocco

\* samiaboussaa@gmail.com

**Data Availability Statement:** All relevant data are within the paper and its Supporting Information files.

## Abstract

Leishmaniasis is endemic in Morocco where both cutaneous and visceral forms coexist. To date, anthroponotic cutaneous leishmaniasis (ACL) determinants remain poorly investigated in Morocco. However, the disease risk factors identification is vital to determine the specific preventive process. In this aim, a case-control study was conducted in the main active ACL foci in central Morocco. Epidemiological data were extracted from bulletins, registers and annual reports of the regional direction of Health offices. The socioeconomic and environmental data were collected from epidemiological surveys, completed by a questionnaire intended for accessible positive population and control people selected from the cases' entourage. The study included 258 cases and 395 controls. Our results showed that many socioeconomic factors were associated with ACL in Morocco such as the rural habitation (OR = 4.163; 95% CI: 2.91–5.96), movement to endemic area (OR = 4.53; 95% CI: 3.03–6.77), provenance from leishmaniasis foci (in Essaouira focus OR = 5.34; 95% CI: 1.19–24.03) and poverty. In addition, environmental factors like proximity of vegetation (OR = 2.45; 95% CI: 1.14–5.25), poor domestic hygiene, particularly the absence of sewage system/waste management (OR = 1.63; 95% CI: 1.35–1.96), and presence of animals (OR = 2.67; 95% CI: 1.14–5.25) increase the risk of ACL in Morocco. Except for Matrimonial status (married people, OR = 4.11; 95% CI: 1.80–9.41), there is however no significant association of the disease with the other socio-demographic factors in the study area (p>0.05). These several risk factors must be taken in consideration to prevent this disease through multidisciplinary collaboration and community participation.

## Introduction

Leishmaniasis is endemic in Morocco where both cutaneous and visceral forms coexist. Human visceral leishmaniasis is caused by *Leishmania* (L) *infantum* MON-1, while cutaneous

**Funding:** The author(s) received no specific funding for this work.

**Competing interests:** The authors have declared that no competing interests exist.

leishmaniasis is caused by three *Leishmania* species. *Leishmania major* MON-1 is responsible for the endemic zoonotic cutaneous form in the southeastern region; *L. infantum* MON-24 is responsible for sporadic cutaneous cases mainly in the northern region; and *L. tropica* causes anthroponotic cutaneous leishmaniasis (ACL) primarily in central Morocco [1, 2].

According to many authors, epidemiological status of visceral leishmaniasis by *L. infantum* and cutaneous leishmaniasis by *L. major* remain comparatively stable in Morocco in terms of geographical distribution [1–3]; while cutaneous leishmaniasis by *L. tropica* continues to appear in non-endemic regions [4, 5]. *Leishmania tropica* notably has the largest geographic distribution in Morocco [6] and it is an etiological cause of human cutaneous leishmaniasis in rural as well as in urban and peri-urban area of Morocco [3, 5]. In addition, Moroccan strains of *L. tropica* show highest genetic diversity with eight zymodems: MON-102, MON-107, MON-109, MON-112, MON-113, MON-122, MON-123 and MON-273 that were reported from human cases, dogs, and the sand fly vector [5, 7].

*Phlebotomus sergenti* is the only proven vector of *L. tropica* in Morocco [6, 7] where it is widespread [8]. The disease is often described as being anthroponotic [7]; however, *L. tropica* was isolated several times from dogs in northern and central Morocco [9–11] and from rodents in central Morocco [12]. Over a period of ten years (2008–2017) 54,838 cases of all forms of leishmaniasis have been reported in Morocco, of which ACL represents 43.3% of all recorded cases. It is characterized by an increasingly widespread dynamic in peri-urban and urban areas with rural character [13].

To date, local ACL determinants remain poorly investigated in Morocco. However, the disease risk factors identification is vital to determine specific preventive methods. In fact, several risk factors have been identified in the spread of leishmaniasis, namely physical and biotic factors, ecosystem diversity, socioeconomic conditions, environmental factors, deforestation due to urbanization, domestic animals and standard of living [14, 15]. Indeed, multiple studies have addressed the effect of seasonality [16], climatic factors [14, 17, 18] and the effect of urbanization on the distribution of leishmaniasis vectors [19–21]. However, in Morocco, studies that address the association between leishmaniasis and human factors are absent. The present study aims to determine the different factors associated with the distribution of ACL in Morocco. Thus, a case-control study was conducted in central Morocco where *L. tropica* ACL still a big health problem.

## Materials and methods

### 1. Study area

**a. Geographical and environmental conditions.** The Marrakech-Safi region lies in central west Morocco (Fig 1) and contains one Prefecture (Marrakech) and seven Provinces (Al Haouz, Chichaoua, El Kelâa Des Sraghna, Essaouira, Rhamna, Safi and Youssoufia), with a total area of 39,167 km$^2$ [22]. The total population of Marrakech-Safi region is 4,520,569 inhabitants, with 43% urban and 57% rural populations, according to High Planning Commission [23].

The 0–500 meters elevation class represents 56.5% of the total area of the Region. Mountain Zone is the mountains of Al Haouz, characterized by high and medium altitudes which rise in the High Atlas range to 4,165 m (Toubkal Mount) [24].

The climate of the region is characterized by apparent variability (average summer temperature of the maxima 37.7C˚ and minima 4.9C˚) with low and irregular rainfall. The rain varies from 800mm in the mountains to 190mm in the plain, influenced by the Atlantic Ocean and the high altitudes of the High Atlas. The arid and semi-arid climates are consistent throughout the region; the sub-humidity appears only in the High Atlas between 1,500 and 2,000 meters.

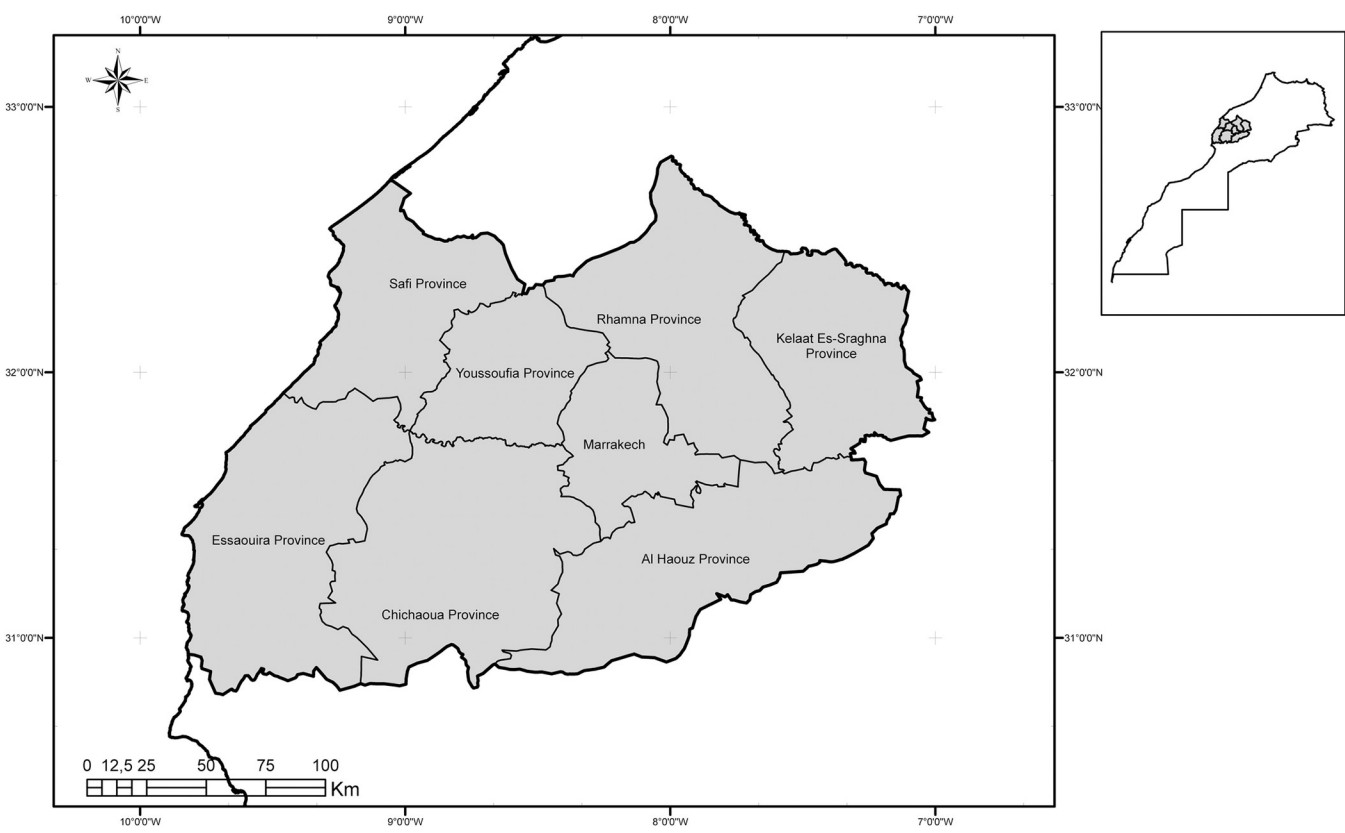

**Fig 1. Presentation of study area [24].**

Almost half of the regional area has rainfall below 300 mm/year [22]. The forests of the Marrakech-Safi region cover an area of 721,876 hectares, representing 22.6% of the national total, including nearly 65,120 hectares of artificial forest and fruit plantations in DRS (Defense and Restoration of Soil). The rate of forest cover in mountain and forest areas is around 43%, with an average of 22% for the entire region. The mountainous character of the Atlas range gives it a pastoral and forestry vocation [23].

**b. Epidemiological status.** Marrakech-Safi region is one of the regions most affected by *L. tropica* cutaneous leishmaniasis (Fig 2). Provinces of Chichaoua, Essaouira, and Al Haouz are the main endemic foci of ACL by *L. tropica* in Morocco. In 2017 they reached respectively 471, 77 and 65 new cases according to Moroccan Ministry of Health [13].

## 2. Study design and population

An unmatched case-control study was conducted to explore the risk factors associated with ACL by *L. tropica* at socio-demographic, socioeconomic and environmental levels. Case population was recruited based on the ACL incidence data, provided by the Epidemiology and Parasitic Disease office of the Regional Health Direction. All patients reported positive for ACL between 2018 and 2020 were eligible if they had resided in the target area for at least one year. The control population was selected from the declared cases' entourage by domicile visit.

The sample size was estimated in OpenEpi (version 3.01) [25], considering the following parameters: Type I error (α) of 0.05 (Using a 95% confidence interval), normal power approximation with continuity correction (1-β) of 90%, minimum detectable OR of 2.00 and 1:1.5 case to control ratio. The 'Fleiss Method with Continuity Correction' was used [26].

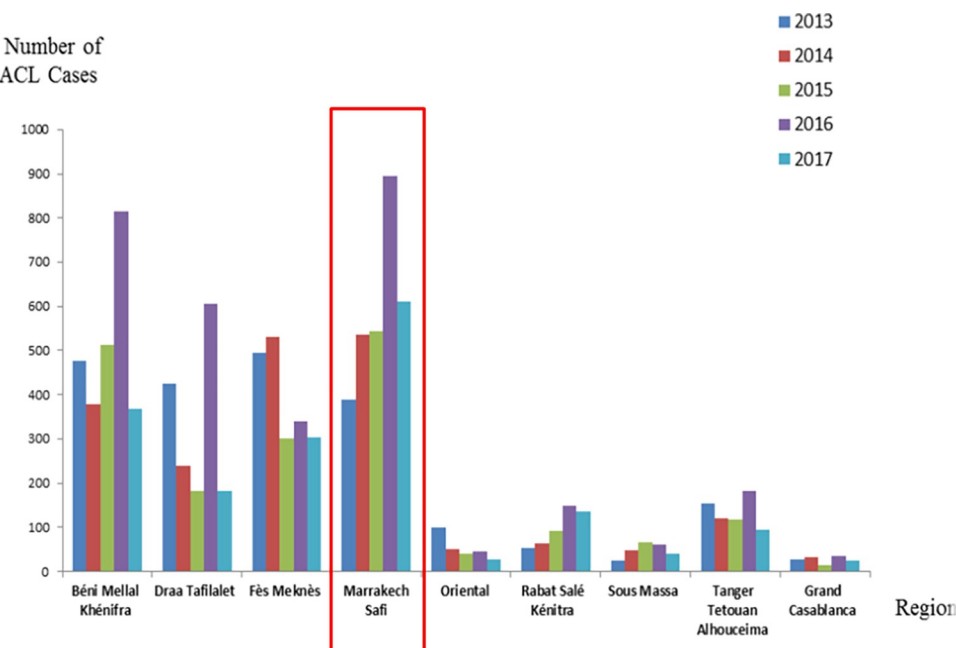

**Fig 2. Distribution of ACL cases in Morocco by endemic region between 2013 and 2017 according to data reported by National Ministry of Health [13].**

Thus, 232 cases and 356 controls were calculated as the minimum number of participants required for this study. However, to account for non-participation and incomplete questionnaires, this number was increased adding 10%. Finally, 258 cases and 395 controls were selected randomly to participate in the present study.

## 3. Variable definition

In the present study, a confirmed case of ACL is a person that presents clinical signs (skin or mucosal lesions) with parasitological confirmation of the diagnosis (positive smear or culture) [27] registered between 2018 and 2020 in Marrakech Safi region.

Control individual is a person without clinical signs, with negative rapid test, negative smear or culture. Controls are recruited from the reports of epidemiological surveys conducted by Health personnel around confirmed cases recorded between 2018 and 2020.

## 4. Data collection

Epidemiological data were extracted from the bulletins, registers and annual reports of the Regional Direction of Health offices. The epidemiological surveys were completed by a structured questionnaire intended for accessible positive population as well as for the control people.

The questionnaire form was designed for an exploratory approach and it covered three sections: socio-demographic (Gender, Age, Matrimonial status, Level of study, Knowledge about leishmaniasis), socio-economic (Number of children per family, Person per household, Provinces, Monthly income, Zone, Travel to endemic area, Profession) and environmental data (Sewage system/waste management, Toilet/Waste water management, Mosquito nets, Vegetation, Farm animals, Companion animals). The categorization was based on previous studies exploring the factors of leishmaniasis in all its forms [28]. The questionnaire form was

validated first by conducting a pre-testing on a small group of local population not included in the final participant list.

### 5. Data analysis

The data were analyzed by using SPSS statistical software (version 21). The association between the dependent and independent variables was measured using the Chi-Square Test (S1 Data). Risk factor data were analyzed by binary logistic regression (S1 Data). Odd Ratio was calculated with a 95% confidence interval for each variable. All statistical tests were performed at a significance level of 0.05 (S1 Data).

### 6. Ethical statement

This study was part of a project approved by the Ethical Hospital-University Committee (Faculty of Medicine and Pharmacy, University of Cadi Ayyad, Marrakech, Morocco) with approval number 020/2016, to carry out an epidemiological study on leishmaniasis in the Marrakech-Safi region. In addition, the official authorization to examine files and data collection was obtained from the Regional Direction of the Moroccan Ministry of Health and Social Protection (Ref.1080/2020). Informed verbal consent was taken from all participants (adults and a parents or guardians of minor participants) before enrolment in the study.

## Results

A total of 653 participants were recruited in our study. 258 (39%) were confirmed cases of ACL caused by *L. tropica* recorded in the region between 2018 and 2020, while 395 (61%) were the correspondent controls (S1 Data).

### 1. Socio-demographic factors

When considering socio-demographic characteristics (Table 1, S1 Data), cases and controls groups were largely represented by the age categories (from 0 to more than 40 years). Females are more affected by the disease (53.49%) and the chances of getting ACL are slightly higher in women (OR = 1.16, 95% CI: 0.84–1.60). The marital status seemed to be associated significantly with ACL ($p$ = 0.005). 51.2% of the cases are single while 67.6% of the controls are married, thus to be married is a risk factor for ACL (OR = 4.11, 95% CI: 1.80–9.41) in the study area.

Regarding population knowledge, the majority of cases and controls had low knowledge about the disease, while exclusively 2.32% had high knowledge about leishmaniasis. No significant correlation was detected between the knowledge variable and ACL, however, a negative correlation was noted between ACL and the level of study (Table 1, S1 Data).

### 2. Social and economic factors

Concerning the socio-economic characteristics (Table 2, S1 Data), most participants in the study are people living in a household of more than three people, but no significant association between household size and ACL disease was found (p = 0.460), unlike the number of children per household. In fact, 77.78% of the cases and 54.55% of the controls have more than two children, while 45.45% of the controls and only 22.22% of the cases have a number of children between 0 and 2. The risk of developing the ACL is higher in families with more than five children (OR = 1.61 95% CI: 0.12–21.39). On the one hand, the association between occupation and ACL is significant statistically (p = 0.056), as almost half of the cases and controls are unemployed, and 34% of the cases and 42.2% of the controls are artisans, farmers, workers or employees. The OR of ACL is the lowest (OR = 0.09) in Civil servants (95% CI: 0.01–0.72). A

**Table 1. Socio-demographic determinants associated with ACL, in the Marrakech-Safi region, Morocco between 2018 and 2020.**

| Variables | Modalities | Case | Control | OR | CI for Exp (B) 95% | | P-value |
|---|---|---|---|---|---|---|---|
| | | N (%) | N (%) | | Lower | Upper | |
| **Gender** | Male Female | 120(46.51) | 175(49.72) | Ref | Ref | Ref | 0.242 |
| | | 38(53.49) | 177(50.28) | 1.16 | 0.84 | 1.60 | |
| **Age (years)** | 0–5 | 43(16.67) | 49(13.92) | Ref | Ref | Ref | |
| | 6–19 | 71(27.52) | 96(27.28) | 1.19 | 0.71 | 1.98 | 0.650 |
| | 20–40 | 87(33.72) | 116(32.95) | 1.19 | 0.72 | 1.94 | |
| | > 40 | 57(22.09) | 91(25.85) | 1.43 | 0.84 | 2.42 | |
| **Matrimonial status** | Single | 62(51.2) | 9(26.5) | Ref | Ref | Ref | |
| | Married | 52(43) | 23(67.6) | 4.11 | 1.80 | 9.41 | |
| | Widowed | 6(4.9) | 1(2.9) | 3.45 | 0.73 | 16.27 | 0.005 |
| | Divorced | 1(0.9) | 1(2.9) | 6.89 | 0.40 | 120.13 | |
| **Level of study** | Literary | 31(23.7) | 9(18) | Ref | Ref | Ref | 0.261 |
| | Primary | 55(42) | 27(54) | 1.69 | 0.71 | 4.05 | |
| | Secondary | 39(29.8) | 14(28) | 1.24 | 0.48 | 3.24 | |
| | Higher level | 6(4.6) | 0(0) | 0.00 | 0.000 | | |
| **Knowledge about leishmaniasis** | Low | 81(62.30) | 42(76.36) | Ref | Ref | Ref | 0.129 |
| | Medium | 46(35.38) | 13(23.64) | 0.545 | 0.265 | 1.119 | |
| | High | 3(2.32%) | 0(0) | 0.000 | 0.000 | | |

**N**: number, **OR**: Odd Ratio, **CI**: Confidence Interval, **Ref**: reference category.

monthly salary of more than 5,000 Dirham can be a protective factor for ACL (OR = 0.06. 95% CI: 0.03–0.14).

The presence of the disease was strongly associated with origin (p≤0.001). Our results confirm that the risk of ACL in Essaouira is 5.34 times higher (95% CI: 1.19–24.03), followed by Marrakech (OR = 1.91) and Chichaoua (OR = 1.28). Among the ACL cases, 57.39% were moved to an endemic area. The majority of controls (77.08%) have no notion of recent travel to endemic area. This showed a strong association between the disease and population displacement (p≤0.001). The chance of acquiring ACL is 4.53 times higher for displaced people (95% CI: 3.03–6.77). Regarding the living environment, 71.60% of ACL cases against only 37.71% of controls were from rural area. A strong association was explored between the residence area and the disease (p≤0.001). The risk of ACL is 4.17 times higher (CI 95%: 2.91–5.96) in rural area compared to the urban setting.

## 3. Environmental factors

Two-thirds (65.39%) of the cases did not have a sanitation and waste management system (Table 3, S1 Data), there is a strong association between sanitation/ waste management system and the ACL disease (p≤0.001). The absence of a sanitation and waste management system tended to increase the risk of ACL 1.63 times (95% CI: 1.35–1.96). However, in this study, there was no significant association between the disease and the provision of toilets and wastewater management.

Inside housing, the majority of cases (72.87%) and majority of controls (83.50%) do not use mosquito nets to protect themselves from sand flies and other insects. In contrast, the use of mosquito nets was considered a risk factor (OR = 2.17; 95% CI 0.58–8.18) in the present study.

Presence of vegetation was also associated with ACL (p = 0.009), most cases (74.80%) and controls (89.70%) had vegetation around their housing. Its absence seemed to increase the risk

**Table 2. Socio-economic determinants associated with ACL, in the Marrakech-Safi region, Morocco between 2018 and 2020.**

| Variables | Modalities | Case | Control | OR | CI for Exp(B) 95% | | P-value |
|---|---|---|---|---|---|---|---|
| | | N (%) | N (%) | | Lower | Upper | |
| **Number of children per family** | < 2 | 12(22.22) | 10(45.45) | Ref | Ref | Ref | |
| | 2 to 5 | 29(53.70) | 5(22.73) | 1.33 | 0.12 | 13.65 | 0.130 |
| | > 5 | 13(24.08) | 7(31.82) | 1.61 | 0.12 | 21.39 | |
| **Person per household** | < 2 | 7(5.56) | 0(0) | Ref | Ref | Ref | 0.460 |
| | 2 to 5 | 63(50) | 7(50) | 0.00 | 0.00 | 0.00 | |
| | > 5 | 56(44.45) | 7(50) | 0.89 | 0.30 | 2.70 | |
| **Province** | Al Haouz | 52(20.16) | 110(27.85) | Ref | Ref | Ref | 0.000 |
| | Chichaoua | 82(31.78) | 26(6.58) | 1.28 | 0.29 | 5.62 | |
| | Essaouira | 21(8.14) | 19(4.81) | 5.34 | 1.19 | 24.03 | |
| | Marrakech | 90(34.88) | 212(53.67) | 1.91 | 0.94 | 9.15 | |
| | Rhamna | 10(3.88) | 23(5.82) | 0.69 | 0.16 | 2.97 | |
| | Youssoufia | 3(1.16) | 5(1.27) | 0.74 | 0.15 | 3.70 | |
| **Monthly income (dhs)** | < 2000 | 57(50.4) | 0(0) | Ref | Ref | Ref | 0.000 |
| | 2000–5000 | 44(38.9) | 11(17.2) | 0.00 | 0.00 | 0.00 | |
| | > 5000 | 12(10.6) | 53(82.8) | 0.06 | 0.03 | 0.14 | |
| **Zone** | Urbain | 73(28.40) | 185(62.29) | Ref | Ref | Ref | 0.000 |
| | Rural | 184(71.60) | 112(37.71) | 4.17 | 2.91 | 6.00 | |
| **Travel to endemic area** | No | 75(42.61) | 232(77.08) | Ref | Ref | Ref | 0.000 |
| | Yes | 101(57.39) | 69(22.92) | 4.53 | 3.03 | 6.77 | |
| **Profession** | C1 | 22(44) | 26(57.8) | Ref | Ref | Ref | 0.019 |
| | C2 | 17(34) | 19(42.2) | 0.68 | 0.03 | 1.57 | |
| | C3 | 7(14) | 0(0) | 0.09 | 0.01 | 0.72 | |
| | C4 | 4(8) | 0(0) | 0.15 | 0.02 | 1.38 | |

**C1**: *No profession*, **C2**: *Craftsmen, farmers, workers, employees...*, **C3**: *Civil servants*, **C4**: *Liberal professions, large traders*, **N**: *number*, **OR**: *Odd ratio*, **CI**: *confidence interval*, **Ref**: reference category, **dhs:** Dirham.

**Table 3. Environmental determinants of ACL in Marrakech-Safi region, Morocco, between 2018 and 2020.**

| Variables | Madalities | Case | | Control | | OR | CI for Exp (B) 95% | | P-value |
|---|---|---|---|---|---|---|---|---|---|
| | | N | % | N | % | | Lower | Upper | |
| **Sewage system/waste management** | Yes | 45 | 34.61 | 187 | 59.74 | 0.58 | 0.45 | 0.75 | 0.000 |
| | No | 85 | 65.39 | 126 | 40.26 | 1.63 | 1.35 | 1.96 | |
| **Toilet/ Waste water management** | Yes | 51 | 46.36 | 185 | 59.10 | 0.93 | 0.77 | 1.11 | 0.222 |
| | No | 59 | 53.64 | 128 | 40.90 | 1.11 | 0.88 | 1.40 | |
| **Mosquito nets** | Presence | 35 | 27.13 | 2 | 12.50 | 2.17 | 0.58 | 8.18 | 0.169 |
| | Absence | 94 | 72.87 | 14 | 83.50 | 0.84 | 0.68 | 1.03 | |
| **Vegetation** | Presence | 95 | 74.80 | 61 | 89.70 | 0.84 | 0.84 | 0.95 | 0.009 |
| | Absence | 32 | 25.20 | 7 | 10.30 | 2.45 | 2.45 | 5.25 | |
| **Farm animals** | Presence | 70 | 55.12 | 6 | 37.50 | 1.47 | 1.47 | 2.82 | 0.143 |
| | Absence | 57 | 44.88 | 10 | 62.50 | 0.72 | 0.72 | 1.10 | |
| **Companion animals** | Presence | 70 | 66.67 | 5 | 25.00 | 2.67 | 2.67 | 5.77 | 0.001 |
| | Absence | 35 | 33.33 | 15 | 75.00 | 0.45 | 0.45 | 0.65 | |

**N**: number, **OR**: Odd ratio, **CI**: confidence interval

of ACL 2.45 times (95% CI 1.14–5.25). Finally, presence of companion animals in close proximity to habitants increase the risk of ACL (OR = 2.67 CI 95%1.14–5.25) (Table 3).

## Discussion

With the aim to detect the main risk factors of ACL in Morocco, many variables were analyzed in the present study. In our results, socioeconomic factors are most significant variables, corroborating with the results of other study that confirm the association of socioeconomic factors with the disease incidence in vulnerable human populations of arid and tropical developing regions [29]. Overcrowded housing attracts sand flies because it is a good source of blood meals [30]. In addition, there is a strong association between the disease and travel to endemic area, which is consistent with other studies showing that the number of cases is increasing according to the increasing number of travelers returning from endemic countries [31]. Regarding occupation, the association with ACL was negatively significant, and the category of civil servants people can be a protective factor for ACL. Generally, outdoor occupations increase the risk of leishmaniasis incidence due to a greater risk of contact with the sand fly vectors [32]. In the other hand, a salary above 5,000 Dirham is a protective factor for ACL, this result is in agreement with a study in Brazil that confirms that the presence of cutaneous leishmaniasis is more important in economically active adults, whose monthly income does not exceed 5,000$ [33]. In the same sense, according to several authors, poverty could be the main determinant of the transmission of leishmaniasis, especially its visceral form [30, 31], as well as the cutaneous form [34]. Conversely, authors in Morocco did not associate the poverty rate with the distribution of cutaneous leishmaniasis [21, 35]. That can be explained by the economic homogeneity of their study population.

The rural environment is associated with a risk of ACL compared to the urban environment. This result is similar to most of the studies conducted in Morocco and elsewhere, on visceral [20], as well as for cutaneous leishmaniasis [34, 36, 37]. The present study shows also that the presence of ACL disease is strongly associated with provenance. The risk comes mainly from Chichaoua and Essaouira, as these Provinces are active foci of cutaneous leishmaniasis [13]. Indeed, the proximity to cases within 50 meters is a risk factor for visceral leishmaniasis [30]. The city of Marrakech may also be at risk of ACL, this result can be explained by urbanization which has a significant effect on the distribution of sand flies the vector and consequently the disease [21, 38]. Certainly, the movement of people from rural to urban area, where housing conditions are unfavorable, contributes to the emergence of the disease [39].

Regarding the environmental factors, our findings show the presence of livestock and the placement of companion animals in close proximity to habitants increase the risk of ACL. This is consistent with previous investigation that confirms an association between cutaneous leishmaniasis and keeping animals at home and in peri-domestic area [33]. The presence of toilets, water pipes and, consequently, a sewage disposal system reduces the incidence of leishmaniasis [33]. Indeed, housing conditions and domestic hygiene are major determinants in the spread of visceral [31] and cutaneous leishmaniasis [40], as they may increase the breeding and rest biotopes of sand flies, as well as their access to humans. Improved housing conditions and personal protection efforts by the poor have the potential to reduce the incidence of visceral leishmaniasis [30]. Inside the home, the majority of local population did not use mosquito nets to be protected from sand flies; this variable was oddly considered as a risk factor. In a study in Pakistan, 63% of infected people with cutaneous leishmaniasis had used mosquito nets [41]. Whereas mosquito nets are a primary form of vector control, especially those impregnated with insecticides [42, 43]. In Iran, several studies confirmed the effect of these strategies for

controlling of zoonotic cutaneous leishmaniasis [44] and ACL [45]. Our result can be explained by adoption of these preventive measures exclusively in the active foci of leishmaniasis, where control program is solicited and implemented by the local Health System [46]. Based on significant association between vegetation and ACL disease, vegetation is considered in our study as a protective factor. Conversely, several studies confirm its positive association with the distribution of cutaneous leishmaniasis [47], mainly natural vegetation [29, 48, 49]. Sand flies are more abundant where vegetation cover and density were greater [50]. The incidence of the disease is associated with a normalized difference in vegetation index (NDVI) [18, 51] and in some cases, deforestation rather reduces human infection [27]. Similarly in Morocco, NDVI showed positive correlation with density of *Phlebotomus sergenti*, the proven vector of ACL [8]. However, others results showed different patterns and magnitudes of the lagged relationship between the vegetation index and cutaneous leishmaniasis incidence [51]. Our results can be explained in one hand by sand fly dispersal, oviposition and feeding behavior. In fact, the results in India showed the mean number of sand flies were significantly higher in banana trees than in other vegetation and in female palmyra palm trees [52]. In other hand, vegetation is not the only factor that influences the incidence of cutaneous leishmaniasis. Other factors, such as temperature, rainfall, and floating population, may also affect the prevalence of the disease [45].

Among the socio-demographic factors only marital status is related to ACL. The risk of developing ACL is four times higher in married than single people in the study area. Although marital status showed significant association with the disease (married people are protected to be affected) in Sri Lanka, it cannot significantly predict the disease incidence [53]. These significant statistic relationships in the univariate analysis are likely to be a result of confounding effects [54], which is not the case of the present study because of the use of regression models [54].

Our results show a predominance of the age category 19–40 years in cases, contrary to other studies that have found the incidence of ACL higher in children [37] between 5 and 14 years [55], and at young age 0–9 years [17, 39]. Generally, the highest prevalence of visceral leishmaniasis is found in children [20, 33, 56], whereas cutaneous leishmaniasis occurs mainly in adult [14, 36, 56]. No association was determined between gender and the disease, corroborating with results of many authors [57, 58]. The preponderance of ACL in females was highlighted [39, 55], however, others confirm a positive association with males in cutaneous cases [14, 33, 36, 37], especially ACL form [33, 37]. Similarly, our study shows no significant association between education level and ACL, while a low level of education is associated with the presence of ACL in other studies [37]. In the same way, we noted no significant association between ACL and the knowledge about the disease, as found by Alhrazi et al, [59]. This result may be attributed to the general lack of awareness and education about ACL disease in the local context. In India, the lack knowledge regarding visceral leishmaniasis was correlated to a low level of education and a low socio-economic status [60].

In sum, sex, education, gender, knowledge, number of children per family, person per family, Toilet/Waste water management, mosquito nets and farm animals were not significantly associated with the incidence of leishmaniasis. However, monthly income, residence zone, travel to endemic area, companion animals, sewage system/waste management and vegetation are potentially linked to ACL. The results suggest that it may be possible to modify a portion of the risk of ACL by making changes in the housing environment which may help to reduce the human–vector contact. Our finding suggests also improving the awareness of the population in endemic and non-endemic areas about the prevention measures, to gain their participation in the fight against the disease through education of local human population.

## Conclusion

This study emphasizes the role of socio-economic and environmental components as the main determinants of ACL incidence in Morocco. Control efforts must be focused on more effective preventive measures that combine the efforts of various sectors. Education, through community participation, thus encourages the population to improve their own hygiene and environmental conditions.

## Supporting information

**S1 Data. Data analysis results.**
(SAV)

## Acknowledgments

Authors thank warmly Jenna Lacey for English revision.

## Author Contributions

**Conceptualization:** Samia Boussaa.

**Data curation:** Mounia Amane, Mohamed Daoudi.

**Formal analysis:** Mounia Amane, Mohamed Echchakery, Mohamed Daoudi, Samia Boussaa.

**Investigation:** Mounia Amane, Mohamed Echchakery.

**Methodology:** Mounia Amane, Mohamed Echchakery, Mohamed Daoudi, Samia Boussaa.

**Software:** Mounia Amane, Mohamed Echchakery.

**Supervision:** Mohamed Echchakery, Samia Boussaa.

**Validation:** Mohamed Hafidi, Samia Boussaa.

**Writing – original draft:** Mounia Amane, Mohamed Echchakery, Mohamed Daoudi.

**Writing – review & editing:** Mohamed Hafidi, Samia Boussaa.

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
