## [Decision Letter · Decision Letter 0]

4 May 2022

PONE-D-22-07886Determinants of Anthroponotic Cutaneous Leishmaniasis by Case-Control Study in MoroccoPLOS ONE

Dear Dr. Boussa,

Thank you for submitting your manuscript to PLOS ONE. After careful consideration, we feel that it has merit but does not fully meet PLOS ONE’s publication criteria as it currently stands. Therefore, we invite you to submit a revised version of the manuscript that addresses the points raised during the review process. Both reviewers acknowledged the relevance of the study in updating aspects of the current knowledge of cutaneous leishmaniasis endemics in Morocco. They also made valuable comments that would help to improve language, distribution of information, and description of the study, as well as to clarify some points in the report. Please, make sure you thoroughly address all issues raised by both reviewers, and attempt to incorporate a substantial proportion of the suggestions in a revised version of the manuscript.

We look forward to receiving your revised manuscript.

Kind regards,

Albert Schriefer, M.D., Ph.D.

Section Editor

PLOS ONE

Journal Requirements:

a)  You may seek permission from the original copyright holder of Figure 1 to publish the content specifically under the CC BY 4.0 license.  

Reviewers' comments:

Reviewer's Responses to Questions

**Comments to the Author**

1. Is the manuscript technically sound, and do the data support the conclusions?

Reviewer #1: Yes

Reviewer #2: Yes

2. Has the statistical analysis been performed appropriately and rigorously? 

Reviewer #1: Yes

Reviewer #2: Yes

3. Have the authors made all data underlying the findings in their manuscript fully available?

Reviewer #1: Yes

Reviewer #2: Yes

4. Is the manuscript presented in an intelligible fashion and written in standard English?

Reviewer #1: No

Reviewer #2: Yes

5. Review Comments to the Author

Reviewer #1: In the current study, some important determinants of local anthroponotic cutaneous leishmaniasis are investigated in Morocco.

Introduction:

Line 47: At the beginning of each sentence the scientific names must be in complete form (L. tropica must be changed to Leishmania tropica).

Results:

There are some errors in table 2 which must be corrected (in Gender line after parenthesis if cases”1” must be deleted. In the line of level of study, “Literary” is misspelled).

Sandfly/sandflies should be written separately (sand fly/ sand flies) in the manuscript.

Line 272: Scientific names must be written in italic form (P. sergenti must be in italic).

Discussion:

Lines 212-223 are not related to the discussion. These sentences can be included in the introduction.

The results should not be repeated in discussion. All p values in the discussion should be omitted. In discussion, the results should be compared with the results of other studies and described.

In the last paragraph of the discussion (lines: 293-301) all factors should be included. Using mosquito nets is missed.

There are many valuable published articles by Iranian scientists. It is recommended that the authors of the current manuscript use them for the introduction and results. For example for the effect of mosquito nets on cutaneous leishmaniasis the following references can be used:

-Moosa-Kazemi SH, Yaghoobi-Ershadi MR, Akhavan AA, Abdoli H, Zahraei-Ramazani AR, Jafari R, Houshmand B, Nadim A, Hosseini M. (2007) Deltamethrin-impregnated bed nets and curtains in an anthroponotic cutaneous leishmaniasis control program in northeastern Iran. Ann Saudi Med. 27(1): 6–12.

- Yaghoobi-Ershadi MR, Moosa-Kazemi SH, Zahraei-Ramazani AR, Jalali-Zand AR, Akhavan AA, Arandian MH, Abdoli H, Houshmand B, Nadim A, Hosseini M. (2006) Evaluation of deltamethrin-impregnated bed nets and curtains for control of zoonotic cutaneous leishmaniasis in a hyperendemic area of Iran. Bull Soc Pathol Exot. 99(1): 43–48.

Reviewer #2: Line 15: "Are" should be replaced with "is".

Line 28: Whether absent of sewage system is beneficial or its presence? its unclear. Please rephrase the sentence.

Line 38: "Are" should be replaced with "is".

Line 54: Here the reference, Boussaa et al is not cited properly, i.e. by numbering. Please do the needful.

Line 99-103: Please explain more about sample size calculation. It is not well explained at all.

Line 119-122: How the data was tabulated and how categorization was done? This part can be more explained for better understanding, like what is the process of categorization etc.

Line 136: Please mention whether the consent taken is written or verbal.

Line 147: Change the phrase as : "females are more affected".

Line 193-195: Sentence is not properly understood. Please re-phrase the sentence.

Line 202: Rephrase the sentence as : presence of vegetation.

Line 224: Use of reference 27 is not understood. You have sated about your study findings here, so how reference 27 is appropriate here?

Line 230: How is this associated? Negatively or positively?

Line 267-275: This part could be better explained, why there is such finding, whats the reason behind it.

Line 278-279: Please explain.

6. PLOS authors have the option to publish the peer review history of their article (what does this mean?). If published, this will include your full peer review and any attached files.

Reviewer #1: **Yes: **Amir Ahmad Akhavan

Reviewer #2: **Yes: **Dr. Moytrey Chatterjee

---

## [Author Response · Author response to Decision Letter 0]

21 Jun 2022

RESPONSE LETTER

PONE-D-22-07886

Determinants of Anthroponotic Cutaneous Leishmaniasis by Case-Control Study in Morocco

PLOS ONE

Dear Editor, 

Thank you for the opportunity to submit a revised version of our manuscript entitled “Determinants of Anthroponotic Cutaneous Leishmaniasis by Case-Control Study in Morocco”. We appreciate the time and efforts you and the reviewers have put into providing your valuable comments on our manuscript. We are grateful for your insightful comments on our paper. We were able to incorporate changes to reflect all the suggestions provided. 

We have highlighted the changes in the manuscript.

Sincerely,

Pr S. BOUSSAA

Corresponding author 

Journal Requirements:

Authors: The revised manuscript was formatted according to PLOS ONE's style 

Authors: The ethics statement section has been rewritten, based on comments from the editor and reviewer, to be more informative.

Authors: The abstract on the online submission was updated.

 a) You may seek permission from the original copyright holder of Figure 1 to publish the content specifically under the CC BY 4.0 license. 

Authors: Figures 1 and 2 were performed by authors; consequently we can publish it on open access form under CC BY 4.0. We used free sheapfile from https://mapcruzin.com/free-morocco-arcgis-maps-shapefiles.htm, for map creation (Fig. 1). 

Figures were presented under TIF format and titles were modified according to the journal recommendations’ 

Reviewer #1: 

In the current study, some important determinants of local anthroponotic cutaneous leishmaniasis are investigated in Morocco.

Introduction:

Line 47: At the beginning of each sentence the scientific names must be in complete form (L. tropica must be changed to Leishmania tropica).

Authors: 

Thank you to the Reviewer for the positive feedback and useful comments. This remark has been corrected throughout the manuscript.

Results:

There are some errors in table 2 which must be corrected (in Gender line after parenthesis if cases”1” must be deleted. In the line of level of study, “Literary” is misspelled).

Authors: Tables were modified according to Reviewer comments 

Sandfly/sandflies should be written separately (sand fly/ sand flies) in the manuscript.

Line 272: Scientific names must be written in italic form (P. sergenti must be in italic).

Authors: These remarks have been corrected throughout the manuscript.

Discussion:

Lines 212-223 are not related to the discussion. These sentences can be included in the introduction.

The results should not be repeated in discussion. All p values in the discussion should be omitted. In discussion, the results should be compared with the results of other studies and described.

In the last paragraph of the discussion (lines: 293-301) all factors should be included. Using mosquito nets is missed.

There are many valuable published articles by Iranian scientists. It is recommended that the authors of the current manuscript use them for the introduction and results. For example for the effect of mosquito nets on cutaneous leishmaniasis the following references can be used:

-Moosa-Kazemi SH, Yaghoobi-Ershadi MR, Akhavan AA, Abdoli H, Zahraei-Ramazani AR, Jafari R, Houshmand B, Nadim A, Hosseini M. (2007) Deltamethrin-impregnated bed nets and curtains in an anthroponotic cutaneous leishmaniasis control program in northeastern Iran. Ann Saudi Med. 27(1): 6–12.

- Yaghoobi-Ershadi MR, Moosa-Kazemi SH, Zahraei-Ramazani AR, Jalali-Zand AR, Akhavan AA, Arandian MH, Abdoli H, Houshmand B, Nadim A, Hosseini M. (2006) Evaluation of deltamethrin-impregnated bed nets and curtains for control of zoonotic cutaneous leishmaniasis in a hyperendemic area of Iran. Bull Soc Pathol Exot. 99(1): 43–48.

Authors: Thank you for these valuable remark and resources. The Discussion section was reviewed according to the comments and all references were included. Please see the revised version of the manuscript 

Reviewer #2: 

Line 15: "Are" should be replaced with "is".

Line 38: "Are" should be replaced with "is".

Authors: Thank you to the Reviewer for the positive feedback and useful comments. These remarks were fixed

Line 28: Whether absent of sewage system is beneficial or its presence? its unclear. Please rephrase the sentence.

Authors: According to our results, the absent of sewage system/waste management increases the risk of ACL which means that its presence is beneficial (protective factor). This sentence was reviewed in the revised version of the manuscript. 

Line 54: Here the reference, Boussaa et al is not cited properly, i.e. by numbering. Please do the needful.

Authors: Thank you for this comment. The manuscript was revised according to the journal style. 

Line 99-103: Please explain more about sample size calculation. It is not well explained at all.

Authors: We added a paragraph in material and methods section about sample size calculation in this case- control study. 

Line 119-122: How the data was tabulated and how categorization was done? This part can be more explained for better understanding, like what is the process of categorization etc.

Authors: We added a paragraph at our opining paragraph of the material and methods section to help the reader to undesrstand : “The questionnaire has three sections covering socio-demographic (Gendre, Age, Matrimonial status, Level of study, knowledge about leishmaniasis), socioeconomic (Number of children per family, Person per household, Provinces, Monthly income (dhs), Zone, Travel to endemic area, Profession) and environmental data (Sewage system/waste management, Toilet/ Waste water management, Mosquito nets, Vegetation, Farm animals, Companion animals). The categorisation was based on previous studies exploring the factors of leishmaniasis in all its forms (28) (29)”. Please see the revised version of the manuscript. 

Line 136: Please mention whether the consent taken is written or verbal.

Authors: The ethics statement section has been rewritten, based on comments from the editor and reviewer, to be more informative.

Line 147: Change the phrase as : "females are more affected". 

Line 202: Rephrase the sentence as : presence of vegetation.

Authors: Phrases were changed accordantly

Line 193-195: Sentence is not properly understood. Please re-phrase the sentence.

Authors: This section was modified according to the Reviewer comments. Please see the revised version of the manuscript. 

Line 224: Use of reference 27 is not understood. You have sated about your study findings here, so how reference 27 is appropriate here?

Authors: This paragraph was rephrased to be clearer. 

Line 230: How is this associated? Negatively or positively? 

Authors: This is associated negatively, we rectified it in manuscript

Line 267-275: This part could be better explained, why there is such finding, whats the reason behind it.

Authors: This part was developed and more explained according to the reviewer comment. 

Line 278-279: Please explain. 

Authors: This paragraph was rephrased to be clearer. Please see the revised version of the manuscript.

---

## [Decision Letter · Decision Letter 1]

13 Jul 2022

PONE-D-22-07886R1Determinants of Anthroponotic Cutaneous Leishmaniasis by Case-Control Study in MoroccoPLOS ONE

Dear Dr. Boussa,

Thank you for submitting your manuscript to PLOS ONE. After careful consideration, we feel that it has merit but does not fully meet PLOS ONE’s publication criteria as it currently stands. Therefore, we invite you to submit a revised version of the manuscript that addresses the points raised during the review process.

More specifically, please, address the minor issue regarding the citations raised by reviewer one. Also, please, kindly carry out one last round of language editing (detailed in the 'Additional Editor Comments' section at the end of this letter) before we can proceed to full acceptance of your manuscript.

We look forward to receiving your revised manuscript.

Kind regards,

Albert Schriefer, M.D., Ph.D.

Section Editor

PLOS ONE

Journal Requirements:

Additional Editor Comments :

Before we can proceed to acceptance, I would like to ask the authors for a last round of English language correction. There are several capitalization and punctuation mistakes affecting mostly the new text inserted to conform with the reviewers suggestions and comments. However, some mistakes are still present in the original text. So, please, make the language review thorough and cautious. The only intent of this demand is to make your study as clear as possible to the general audience, potentially increasing its reach.

Reviewers' comments:

Reviewer's Responses to Questions

**Comments to the Author**

1. If the authors have adequately addressed your comments raised in a previous round of review and you feel that this manuscript is now acceptable for publication, you may indicate that here to bypass the “Comments to the Author” section, enter your conflict of interest statement in the “Confidential to Editor” section, and submit your "Accept" recommendation.

Reviewer #1: (No Response)

Reviewer #2: All comments have been addressed

2. Is the manuscript technically sound, and do the data support the conclusions?

Reviewer #1: Yes

Reviewer #2: Yes

3. Has the statistical analysis been performed appropriately and rigorously? 

Reviewer #1: Yes

Reviewer #2: Yes

4. Have the authors made all data underlying the findings in their manuscript fully available?

Reviewer #1: Yes

Reviewer #2: Yes

5. Is the manuscript presented in an intelligible fashion and written in standard English?

Reviewer #1: Yes

Reviewer #2: Yes

6. Review Comments to the Author

Reviewer #1: Thanks for considering my comments BUT some references in the text are not matched with the number indicated in the References section such as the references 45-47. Please check all references throughout the manuscript and match them with the corresponding number in the References section.

Reviewer #2: (No Response)

7. PLOS authors have the option to publish the peer review history of their article (what does this mean?). If published, this will include your full peer review and any attached files.

Reviewer #1: **Yes: **Amir Ahmad Akhavan, Professor of Medical Entomology and Vector Control, School of Public Health, Tehran University of Medical Sciences, Tehran, Iran

Reviewer #2: **Yes: **Dr. Moytrey Chatterjee

---

## [Editor Report · Decision Letter 2]

9 Sep 2022

PONE-D-22-07886R2Determinants of Anthroponotic Cutaneous Leishmaniasis by Case-Control Study in MoroccoPLOS ONE

Dear Dr. Boussa,

Thank you for submitting your manuscript to PLOS ONE. After careful consideration, we feel that it has merit but does not fully meet PLOS ONE’s publication criteria as it currently stands. Therefore, we invite you to submit a revised version of the manuscript that addresses the points raised during the review process.

More specifically, please, kindly make the very minor edits suggested in the 'Additional Editor Comments' section below before we move on to acceptance of the manuscript.

We look forward to receiving your revised manuscript.

Kind regards,

Albert Schriefer, M.D., Ph.D.

Section Editor

PLOS ONE

Journal Requirements:

Additional Editor Comments (if provided):

Line 130, Methods. Please adjust the sentence 'a confirmed case of ACL is a person presents clinical signs' to read 'a confirmed case of ACL is a person that presents clinical signs'.

Line 208, Results. Please adjust the sentence 'The chance of acquiring ACL is 4.53 times more increased for displaced people' to read 'The chance of acquiring ACL is 4.53 times higher for displaced people'.

Line 234, Results. Please check if the word 'habitants' would not be the correct one, instead of 'habitats' in this sentence.

Line 249, Discussion. Please, adjust the sentence 'that category of civil servants people can be a protective factor for ACL' to read 'and the category of civil servants people can be a protective factor for ACL'.

Line 271, Discussion. Please check if the word 'habitants' would not be the correct one, instead of 'habitats' in this sentence.

Line 283, Discussion. Please, adjust the sentence 'several studies confirmed the effect of this strategies for control of zoonotic cutaneous leishmaniasis' to read 'several studies confirmed the effect of these strategies for controling of zoonotic cutaneous leishmaniasis'.

Line 330, Discussion. Please, adjust the sentence 'Our finding suggests also improving the awareness of the population in endemic and non-endemic area' to read 'Our finding suggests also improving the awareness of the population in endemic and non-endemic areas'.

---

## [Editor Report · Decision Letter 3]

26 Sep 2022

Determinants of Anthroponotic Cutaneous Leishmaniasis by Case-Control Study in Morocco

PONE-D-22-07886R3

Dear Dr. Boussa,

We’re pleased to inform you that your manuscript has been judged scientifically suitable for publication and will be formally accepted for publication once it meets all outstanding technical requirements.

Kind regards,

Albert Schriefer, M.D., Ph.D.

Section Editor

PLOS ONE
---

## [Editor Report · Acceptance letter]

5 Oct 2022

PONE-D-22-07886R3 

DETERMINANTS OF ANTHROPONOTIC CUTANEOUS LEISHMANIASIS BY CASE-CONTROL STUDY IN MOROCCO 

Dear Dr. Boussaa:

I'm pleased to inform you that your manuscript has been deemed suitable for publication in PLOS ONE. Congratulations! Your manuscript is now with our production department. 

Kind regards, 

on behalf of

Dr. Albert Schriefer 

Section Editor

PLOS ONE